# Immobilization of Radionuclide ^133^Cs by Magnesium Silicate Hydrate Cement

**DOI:** 10.3390/ma13010146

**Published:** 2019-12-30

**Authors:** Tingting Zhang, Tong Li, Jing Zou, Yimiao Li, Shiwei Zhi, Yuan Jia, Christopher R. Cheeseman

**Affiliations:** 1Faculty of Infrastructure Engineering, Dalian University of Technology, Dalian 116023, China; tingtingzhang@dlut.edu.cn (T.Z.); tongtongli@mail.dlut.edu.cn (T.L.); zou_jing8@mail.dlut.edu.cn (J.Z.); lym_z@mail.dlut.edu.cn (Y.L.); dlutzsw@mail.dlut.edu.cn (S.Z.); 2Hebei Provincial Key Laboratory of Inorganic Nonmetallic, College of Materials Science and Engineering, North China University of Science and Technology, Tangshan 063210, China; 3Department of Civil and Environmental Engineering, Imperial College London, London SW7 2AZ, UK; c.cheeseman@imperial.ac.uk

**Keywords:** cesium ions (Cs^+^), immobilization, leaching, magnesium silicate hydrate cement, nuclide

## Abstract

The radionuclide cesium (Cs) was solidified using magnesium silicate hydrate (M–S–H) cement. The influence of Cs^+^ on the reaction of the M–S–H gel system was evaluated by measuring the compressive strength and microscopic properties of the solidified body. By testing the impact resistance, leaching resistance and freeze–thaw resistance of the solidified body, the immobilizing ability of Cs^+^ by the M–S–H cement was analyzed. Results indicate that Cs^+^ only slightly affects the reaction process of the M–S–H gel system, and only slows down the transformation rate of Mg(OH)_2_ into the M–S–H gel to a certain extent. The M–S–H cement exhibits superior performance in solidifying Cs^+^. Both the leaching rate and cumulative leach fraction at 42 d were considerably lower than the national requirements and better than the ordinary Portland cement-solidified body. The curing effect of the M–S–H cement on Cs^+^ is mainly physical encapsulation and chemisorption of hydration products.

## 1. Introduction

Nuclear energy has been widely used as a clean and efficient energy resource. Despite the considerable economic and social benefits of nuclear energy, the problem concerning the large amount of nuclear waste to be disposed of requires global attention [1,2,3]. A total of 360,000 tons of radioactive nuclear waste are accumulated by 445 nuclear reactors, with an annual growth rate of 12,000 tons. Radioactive waste liquid is mostly produced during power generation [4,5,6]. Cs is one of the most common and most harmful radionuclides in low and intermediate radioactive solid wastes. A total of 34 cesium isotopes have been found, including ^137^Cs and ^134^Cs. Among these isotopes, cesium-133 is the only stable isotope found in nature, and the remaining radioactive isotopes are produced by uranium fission [7]. ^137^Cs has a long half-life of 30.5; moreover, active Cs exhibits medium and high-level radiotoxicity. ^137^Cs is highly active and typically exists in the form of ions. Moreover, it can easily migrate in soil and water and cause radioactive pollution when entering an environment with groundwater [8,9,10]. Solidification is an important process of radioactive waste disposal. In experiments, inert Cs (e.g., CsCl) instead of active Cs is often used to avoid radiotoxicity [6,7].

The most commonly used immobilization matrices for low- and intermediate-level wastes are currently based on ordinary Portland cement (OPC). Modified OPC is used to solidify and then store radioactive wastes in steel barrels [11,12,13,14,15,16]. It provides a stable solidification performance, involves a simple process and is cost-efficient. However, OPC itself as a cement curing material has several disadvantages, such as poor durability and high porosity [17,18,19,20]. Problems arise during solidification, and these include the low capacity of the nuclear waste package, apparent increase in solidification capacity, and ease of nuclide migration. The adsorption rate of Cs^+^ is less than 0.3% of the original concentration [21].

Consequently, numerous studies use new gelled material systems, such as alkali-activated materials, phosphate cement and hydrothermal synthetic materials, among others, to solidify radioactive waste [22,23,24,25]. Vandevenne et al. [26] investigated the curing of Cs^+^ by alkali-activated materials with different Ca–Si–Al ratios. Cs^+^ immobilization was proven to be higher at lower Si/Al and Ca/(Si + Al) ratios. Lai et al. [27] examined the curing properties of magnesium phosphate cement on Cs^+^ and Sr^2+^, and they found that the solidification of Cs^+^ by magnesium phosphate cement was mainly based on chemical bonding, supplemented by the physical encapsulation of cement-hydrate to prevent nuclide leaching. The immobilization potential of alkaline-activated fly ash matrices for Cs^+^ with NaOH solution as the activator was examined by Fernandez-Jimenez et al. [28]. Moreover, Cs^+^ was shown to exert no significant adverse effects on mechanical strength or microstructure, and was incorporated into the alkaline aluminosilicate gel. Different siliceous minerals with excellent adsorption properties were applied to solidify partly evaporated bottoms from nuclear power plants to improve the efficiency of solidification [29].

The M–S–H gel system has a porous structure with a specific surface area of about 200 m^2^/g. Therefore, it possesses excellent adsorption, particularly for Cu, Ni, and other heavy metal ions. The M–S–H gel system has a low and stable [30] heat of reaction, which is favorable for reducing the migration ability and improving the curing ability of activated ions in nuclear waste. Compared with the OPC-solidified body, the M–S–H cement-solidified body has a lower porosity, thereby ensuring the leaching rate and strength. In addition, the stable pH of the M–S–H gel system ranges from 9 to 10, which is beneficial for the direct sealing of nuclear wastes containing aluminum and other metals. The experimental results, based on the amount of hydrogen produced during the reaction between active metal and solidified matrix, indicated that the production of H_2_ is considerably lower than that of OPC and geopolymers [31,32]. All these characteristics show that the M–S–H system can potentially solidify nuclear waste.

In the present study, non-radioactive ^133^Cs was used to replace ^137^Cs, which is the most common and harmful radionuclide in a low- and intermediate-level radioactive waste form, and solidified in the new low-alkalinity M–S–H gel system. This study aims to establish whether the hydration reaction of the M–S–H gel system changes with the addition of Cs^+^, and to clearly determine the leaching effect of the M–S–H cement on Cs^+^, compared with that of the OPC system.

## 2. Materials and Methods

### 2.1. Materials

The materials and chemicals used to prepare the cement mixtures were light-burned magnesium oxide (MgO, MagChem 30; MgO activity of 88.40%, M.A.F. Magnesite B.V., City, The Netherlands), silica fume (SF, Elkem Materials Ltd., Shanghai, China), silica sand (Xinlian Quartz Ore, China, >98% SiO_2_, Xinlian Quartz Sand Mine, Liaoning, China), sodium hexametaphosphate (NaHMP, Sinopharm Group, Shanghai, China), cesium chloride (CsCl, Sinopharm Group, Shanghai, China) and ordinary Portland cement (OPC, P.O 42.5R, Dalian Onoda Cement Co. Ltd., Dalian, China). The characteristics of the raw materials are listed in Table 1 and Figure 1. The mean particle size of the as-received sand was 168.84 μm.

### 2.2. Methods

#### 2.2.1. Preparation of Samples

A previous study [32,33,34] has shown that when the atomic Mg/Si ratio is within the 0.67–1.0 range, MgO fully reacts with silica fume (SF) to generate the most M–S–H gels. Thus, an atomic Mg/Si ratio of 1—that is, a blend of 60 wt % SF and 40 wt % MgO—was investigated to prepare hydrated M–S–H cement. Results obtained in earlier research [32,35] indicated that the water-reducing effect of the system is optimal when the content of sodium hexametaphosphate is 2 wt % of the M–S–H system. Therefore, this parameter was used in the present study. The sand ratio (the weight of sand to the total weight of MgO and SF) was 50%. The content of ^137^Cs in the actual radioactive waste was less than 1 wt %; accordingly, the contents of ^133^Cs (calculated using the mass of Cs) were selected as 0.4 wt %, 0.8 wt % and 1.6 wt % of the total weight of MgO and SF. The water–cement ratio was set at 0.7. Before each sample was prepared, a specific amount of NaHMP was dissolved in water, and CsCl was added into the mixture. Subsequently, 40% MgO was mixed with 60% SF by weight as powder and added into water three times. The mixture was blended using a cement-and-mortar mixer for 90 s at a low speed and then for 90 s at a high speed each time.

The paste was held on a vibration table for 5 min to remove air bubbles after mixing. For mechanical testing, the samples were prepared by casting them in steel molds to produce 20 mm × 20 mm × 20 mm samples. These samples were cured in sealed boxes with a relative humidity of >98% to protect them from drying out during curing. The samples were removed from the molds after 24 h and taken out of the boxes after 1, 3, 7, 28 and 90 d. To test for leaching resistance, freeze–thaw resistance and impact resistance, the samples were cast into molds to produce Φ50 × 50 mm samples. For the comparison test, the water–cement ratio of OPC was set at 0.4; the nuclide content, sample size and curing method were consistent with the M–S–H cement-solidified body.

#### 2.2.2. Mechanical Strength

Compression tests were conducted using a microcomputer control electronic universal testing machine (WDW-50, Changchun New Testing Machine Co., Ltd., Changchun, China) at a loading speed of 0.5 mm/min.

#### 2.2.3. Microstructural Analysis

X-ray diffraction (XRD, D8 Advance with CuKα, Bruker, Karlsruhe, Germany) was conducted to identify the crystalline phases present in the samples. Experiments were performed in the 2θ range between 5° and 80°. Fourier-transform infrared spectroscopy (FTIR, EQUINOX55, Bruker, Karlsruhe, Germany) was used to determine the chemical bonds and functional groups of the hydration products. The microstructure of these hydration products was analyzed by scanning electron microscopy (SEM, Nova NanoSEM-50, FEI Company, Hillsboro, OR, USA). The pore structures of the samples were analyzed by mercury intrusion porosimetry (MIP, AutoPore IV9500, McMurray Instruments, Atlanta, GA, USA).

#### 2.2.4. Leaching Resistance

In accordance with the requirements for the standard test for the leachability of low and intermediate-level solidified radioactive waste forms (GB/T 7023-2011), and the performance requirements for low and intermediate level radioactive waste form–cemented waste forms (GB14569.1-2011), the leaching performance of the solidified body composed of the M–S–H cement was evaluated.

The preparation and testing methods are described as follows: After curing for 28 d, the block face was polished with sandpaper. Floating dust particles were cleaned and then collected into a polyethylene bottle for testing. The waste matrices were immersed in a polyethylene bottle containing distilled water or simulated sea water (Table 2) at ambient temperature (25 °C or 40 °C). The leaching solution was replaced from the beginning of the leaching test (1, 3, 7, 10, 14, 21, 28, 35, 42 and 70 d). The concentration of Cs+ in the leaching solution was measured by inductively coupled plasma spectrometry (ICP, Optima 2000 DV, PerkinElmer, MA, USA). The nuclide leaching rate and cumulative leach fraction (CLF) were calculated.

The leaching rate (*R_n_*) and CLF (*P_t_*) were calculated using the following equation:
(1)Rn=anA0(SV)·(Δt)n
(2)Pt=∑anA0SV
where *a_n_* is the mass of component *i* in the *n*th leaching period (g), *A*_0_ is the initial mass of component *i* in the specimen (g), *S* is the exposed surface area of the specimen (cm^2^), *V* is the volume of the specimen (cm^3^), (∆*t*)*_n_* is the last day of the nth leaching period (d) and *t* is the total number of leaching days (*t* = *∑*(∆*t*)*_n_*,*d*).

#### 2.2.5. Other Performances

(1) Impact resistance

The test block was subjected to a vertical free fall from 9 m high to the concrete floor, and was observed to assess whether the solidified body was evidently broken (small angular fragments and cracks do not render the body broken).

(2) Freeze–thaw resistance

The solidified body cured for 28 d was subjected to freeze–thaw testing. When the temperature in the freezer reached 15 °C, the samples packed in airtight plastic bags were placed into the freezer. When the temperature dropped to −15 °C, the freezing time was calculated. The duration of freezing was 3 h (the freezing ranged from 20 °C to −15 °C). After freezing, the samples (together with the plastic bag) were removed and immediately allowed to dissolve in a 15 °C–20 °C water tank; the dissolution time was 4 h. Five freeze–thaw cycles were performed for each sample to observe its appearance and determine its compressive strength and quality.

## 3. Results

### 3.1. Mechanical Strength

The compressive strengths of the MgO/SF-based mortar with different contents of Cs^+^ after 3, 7, 28 and 90 d is shown in Figure 2. The strength development of the M–S–H cement is markedly slow in the early stages, and the compressive strength is relatively low. The compressive strength of the samples without Cs^+^ significantly increases with an increase in age and exceeds 55 MPa until 90 d. This strength is attributed to the porous and low-strength Mg(OH)_2_ produced at the early hydration stage. As hydration progresses, the Mg(OH)_2_ gradually reacts with the SF and a denser M–S–H gel is formed over time.

After the addition of Cs^+^, the compressive strength in each curing stage decreases to varying degrees. With an increase in the addition of Cs^+^, the rate of decline of compressive strength increases, the compressive strength of the solidified body with 0.4 wt %, 0.8 wt % and 1.6 wt % Cs^+^ after 90 d decreases by 21.2%, 41.2% and 49.46%, respectively. With an increase in age, the compressive strength of the solidified body with 1.6 wt % Cs^+^ after 3, 7, 28 and 90 d decreases by 0.24, 5.88, 22.07 and 27.72 MPa, respectively. The compressive strength of the solidified body with 1.6 wt % Cs^+^ for 7 d reaches 12.52 and 28.33 MPa at 90 d, which are considerably higher (about four times) than the 7 MPa required by the national standard GB14569.1-2011. The decrease in strength of the cementitious system is mainly attributed to the loosening of the structure when Cs^+^ is added Cs^+^ inhibits the development of the compressive strength of the samples.

### 3.2. XRD Characterization

Figure 3 presents the X-ray diffraction data of the samples with 1.6 wt % Cs^+^ after 3, 7, 28 and 90 d. In the first 7 d, the reaction products are Mg(OH)_2_ and M–S–H gel (2θ at approximately 35° and 60°). After 7 d, the intensities of the MgO peaks at 43° and 62° decreases with time. The peaks of brucite also decrease with time, indicating that brucite continues to react with SF to form the M–S–H gel. At 90 d, most hydration products are M–S–H gels.

Figure 4 shows the XRD data of the samples with 1.6 wt % Cs^+^ and those without Cs^+^ after curing 28 and 90 d. The amount of Mg(OH)_2_ in the sample with Cs^+^ is more than that in the blank sample at 28 and 90 d, indicating that the addition of Cs^+^ reduces the rate of transformation from Mg(OH)_2_ to the M–S–H gel. In addition, no new substance is found.

### 3.3. Fourier-transform Infrared Spectroscopy Characterization

Corresponding Fourier-transform infrared spectroscopy (FTIR) data of the silica fume and M–S–H cement-solidified body with 1.6 wt % Cs^+^ after curing 3, 7, 28 and 90 d are presented in Figure 5. The absorption peaks of active SiO_2_ in SF are mainly around 473, 808 and 1117 cm^−1^, which are the characteristic peaks of tetra-coordinated silicon atoms. The sharp absorbance near 3690 cm^−1^ is a characteristic Mg–OH stretch. The weak absorption peak appearing at 898 cm^−1^ is associated with Si–OH stretching vibrations. The bands at 1007 and 1068 cm^−1^ correspond to the Si–O–Mg asymmetric stretching vibration peak. Owing to the complex structure of the M–S–H gel, different bonding modes of the Si–O–Mg chemical bond, or changes in the Si–O–Mg bond angle, the asymmetric splits. The band appearing at 550–640 cm^−1^ corresponds to Si–O–Mg bending vibrations.

With the continuous hydration reaction, the absorption peak of the symmetric Si–O stretching vibration (808 cm^−1^) gradually disappears, and the absorption peak of the asymmetric Si–O stretching vibration (1117 cm^−1^) gradually shifts to 1007 and 1068 cm^−1^. This result indicates the decomposition of SiO_2_ and the formation of new siliceous phases. The position of the absorption peak (550–640 cm^−1^) proves the formation of the Si–O–Mg bond. A large amount of Mg(OH)_2_ is generated in the initial hydration stages of MgO, but gradually disappears after 28 d, indicating the consumption of Mg(OH)_2_ by the reaction.

As presented in Figure 6, the addition of Cs^+^ changes the proportion of the asymmetric stretching vibration peaks (1007 and 1068 cm^−1^) of the two Si–O–Mg chemical bonds, which may change the bonding mode of the Si–O–Mg chemical bond or the Si–O–Mg bond angle. In addition, when Cs^+^ is added, the types of hydration products are hardly affected, and no new characteristic peaks are found.

### 3.4. Scanning Electron Microscopy

Figure 7 shows the SEM images of the M–S–H cement with 1.6 wt % Cs^+^ at different curing ages. Analysis indicates that after curing for 3 d, the sample has many pores, and the reaction products exhibit accumulation and adhesion by blocky particles. The pore size of the Mg(OH)_2_ sheet is approximately 1.8 microns. After curing for 7 d, a large number of flocculent aggregates appear in the samples, representing M–S–H gels overlapping with one another. The Mg(OH)_2_ morphologies show a triangular pyramid with a length of about 0.4 microns. Mg(OH)_2_ is consumed during the M–S–H gel formation; thus, Mg(OH)_2_ exhibits dissociation and serration, leading to the formation of a lamellar structure. After curing for 28 d, the pores decrease, and Mg(OH)_2_ presents a three-way cone with holes, embedded in the M–S–H gel. After curing for 90 d, the unreacted SF particles are significantly reduced and the pores are further reduced, forming a large number of gel-like substances; meanwhile Mg(OH)_2_ crystals disappear. This process is basically consistent with the reaction of the M–S–H cement, indicating that the addition of Cs^+^ only slightly influences the reaction process of the M–S–H cement. Surface scanning (EDS) analysis of the samples in Figure 7c was performed, and the elemental distribution is shown in Table 3. The atomic molar ratios of Mg and Si in regions A, B and C were 0.83, 0.92 and 0.81, respectively, and should be the M–S–H gel (Table 3). The presence of the Cs nuclide can be detected in the three positions. The weight percentage is only slightly lower in the B area, indicating that Cs can be more uniformly solidified in the hydration products.

Figure 8 shows the microstructure of hydrated products with different contents of Cs^+^ at 28 d. When the content of Cs^+^ is relatively small, no significant change in the microstructure and compact degree is observed, and the reaction product is mainly M–S–H gel. With an increase in Cs^+^ content, the degree of compactness of the sample decreases significantly.

### 3.5. Leaching Results

Table 4 and Figure 9 present the leaching rate and CLF of the solidified body with 0.8 wt % Cs^+^ in deionized water and simulated sea water. The results show that the leaching rate decreases with an increase in the leaching time. In the early stages of the immersion test, the leaching rate of the solidified body decreases significantly, and the rate of decline markedly slows down after 10 d; the leaching rate then tends to be flat. When the soaking cycle reaches 42 d, the leaching rate of Cs^+^ from the M–S–H cement-solidified body under the two leaching agents is only 5.34 × 10^−5^ cm/d and 4.69 × 10^−4^ cm/d, both of which are lower than the leaching rate limit of nuclide at 42 d, as stipulated in GB14569.1-2011 (Cs^+^ < 4 × 10^−3^ cm/d). The CLFs of the solidified body are 8.89 × 10^−3^ and 5.60 × 10^−2^ cm, both of which are considerably less than the required 42 d stipulated in GB14569.1-2011 (Cs^+^ < 0.26 cm), which are only 3.42% and 21.54% of the standard requirement.

In addition, the leaching rate of the sample with simulated sea water is greater than that with deionized water in each leaching cycle; however, the leaching rate of the solidified body with simulated sea water decreases significantly, and the difference in the Cs^+^ leaching rate in the solidified body gradually narrows under the two leaching fluids. The CLF markedly increases before 10 d, but increases slowly in the later stages of leaching and tends to be flat. The CLF of deionized water is lower, the change trend in the late stages is slower, and the effect of curing is better.

Table 5 and Figure 10 show the comparison results of the leaching rate and CLF between the M–S–H cement-solidified body and the OPC-solidified body mixed with 0.8 wt % Cs^+^ in deionized water. At the beginning of the immersion time, the leaching rate of the OPC-solidified body is much higher than that of the M–S–H cement-solidified body. At the end of the immersion time, the Cs^+^ leaching rate gradually levels off, and the gap decreases.

The CLF of the OPC-solidified body is much higher (6–9 times) than that of the M–S–H cement-solidified body. Moreover, the CLF of the OPC-solidified body largely varies from that of the M–S–H cement-solidified body. The CLF of the M–S–H cement-solidified body is consistently low, indicating that the effect of curing Cs^+^ by the M–S–H cement is better.

### 3.6. Other Performances

#### 3.6.1. Impact Resistance

The test results for the M–S–H cement-solidified body with 0.8 wt % Cs^+^ after anti-shock testing are presented in Figure 11. Small fragments appear in contact with the ground. The whole test block is not broken, and its integrity is still maintained. The results show that the cement waste form with the M–S–H cement as the solidified base material exhibits good impact resistance, which meets the requirements of GB14569.1-2011.

#### 3.6.2. Freeze–thaw Resistance

The changes in compressive strength and mass losses in the M–S–H cement-solidified block before and after freeze–thaw testing are listed in Table 6. After five freeze–thaw cycles, the strength decreases by 13.97%. The weight hardly changes before and after testing, and the mass loss rate is only 0.52%.

These results meet the requirements of GB14569.1-2011 that the compressive strength loss before and after the freeze–thaw cycle shall not exceed 25% and the quality loss shall not exceed 5%.

Figure 12 shows the apparent morphology of the samples before and after the freeze–thaw test. After five freeze–thaw cycles, no expansion or crack in the appearance of the test block is observed, indicating that the M–S–H cement mixed with Cs^+^ exhibits satisfactory freeze–thaw resistance.

## 4. Discussion

The leaching performance of Cs^+^ in the solidified body is the most important performance to evaluate whether the solidified matrix can be used for curing. The method for curing Cs^+^ in the M–S–H cement is discussed based on the leaching results as well as the microanalysis of solidified bodies and other property analyses.

When the water–cement ratio is high, the M–S–H cement-solidified body still maintains good compressive strength and a relatively dense internal structure. Analysis by XRD, IR, and SEM indicates that the addition of Cs^+^ only slightly influences the reaction of the M–S–H cement, and only slows down the conversion of Mg(OH)_2_ into the M–S–H gel to a certain extent. The excellent mechanical properties of the solidified body are ensured.

Cs^+^ in the solidified body is partly bound (adsorbed or solid solution) in the solid phase and partly in the pore solution in a free state. The concentration of Cs^+^ in the pore solution is directly related to the binding capacity of the substrate of the solidified body. The initial amount of Cs^+^ in the pore solution is proportional to the porosity. When the solidified body is immersed in the leaching agent, Cs^+^ bound on the surface of the solid phase is desorbed or dissolved. Simultaneously, liquid–liquid diffusion occurs between the solution in the connected hole and leaching agent, and the solid phase of the leaching agent, solution pore, wall of hole, tends to be balanced. Although the solution in the closed hole is isolated from the connected hole, it should also be in equilibrium with the solid phase in contact with it. Therefore, Cs^+^ in the closed hole solution needs to undergo solid-phase diffusion to reach the connected hole or the surface of the solidified body [36].

Figure 13 shows the mercury injection test results of the M–S–H cement-solidified body and OPC-solidified body. Table 7 lists the pore structure analysis of two cement-solidified bodies. The results show that the amount of mercury in the OPC-solidified body reaches the maximum at the pore size of 62.59 nm, and the pore size is concentrated between 10 and 100 nm. Most pores in the M–S–H cement-solidified body are small holes with an aperture less than 10 nm. The OPC-solidified body pores have a pore size about 90,000 nm more than the M–S–H cement-solidified body. The average pore diameter and median pore diameter are higher than those of the M–S–H cement-solidified body. The total pore area of the M–S–H cement-solidified body is 35.72 m^2^/g, the porosity is 24.98% and the total pore area of the OPC-solidified body is 20.61 m^2^/g; the porosity is 32.27%, which is higher than that of the M–S–H cement-solidified body. The OPC-solidified body has numerous large pores, and the pores with the largest pore size range from 10 to 100 nm; however, the M–S–H cement-solidified body has the largest proportion of pores smaller than 10 nm.

Comprehensive analysis indicates that the porosity of the M–S–H cement-solidified body is lower than that of the OPC-solidified body; moreover, a small proportion of the M–S–H cement solidified body has a pore diameter larger than 10 nm, whereas the majority of the OPC-solidified body has pore diameters larger than 10 nm. The M–S–H cement-solidified body has a large total pore area and consists of numerous tiny pores, indicating that the hydration product has a large specific surface area; in addition, the adsorbable ion content is high. The pore size distribution also reveals that ions diffusion through a pore solution in the solidified body exhibits high resistance, and Cs^+^ is largely hindered by the M–S–H cement hydration products; thus, the nuclides leaching rate is lower.

Solubility directly affects the concentration of ions in the pore solution and the leaching rate; the higher the solubility, the higher the leaching rate. The solubility of Mg(OH)_2_ in the M–S–H cement hydration product is 0.009 g/L (18 °C). The M–S–H gel exhibits a lower solubility, and the solubility of OPC hydration products ranges from 0.5 g/L to 1.3 g/L [32]. The solubility of the M–S–H gel is lower than the OPC hydration product; Cs^+^ in the pore solution of the M–S–H cement exhibits a lower concentration, further reducing the leaching rate.

Therefore, the Cs^+^ adsorbed in the hydration product is not easily released into the pore solution with the dissolution of the hydration product, which facilitates the reduction of the leaching rate.

The M–S–H cement exhibits a low and stable heat of reaction [30], reducing the migration ability of activated ions and the leaching rate, and further improves the curing ability.

The binding capacity (solid solution or adsorption) of the substrate and its hydration products to the nuclides is also key to determining the leaching rate. MgO and SiO_2_ are raw materials of the M–S–H cement with a large specific surface area. The main reaction products of the M–S–H cement are Mg(OH)_2_ and M–S–H gel. M–S–H gel is an amorphous substance with a large specific surface area (200 m^2^/g). It shows excellent adsorption performance for Cs^+^ and can contain a certain amount of heavy metal elements [33,34]. However, the crystalline products after the hydration of OPC-calcium hydroxide and calcium sulfoaluminate hydrates—exhibits almost no ion adsorption capacity. High ion adsorption capacity is also an important reason for the lower leaching rate of the M–S–H cement-solidified body that of the OPC solidified body.

In summary, the M–S–H cement has a strong curing ability for Cs^+^ and exerts two effects on curing Cs^+^: physical encapsulation and adsorption of hydration products. The M–S–H cement is used to solidify radionuclides. The leaching rate of radionuclides is much lower than that of the OPC system. Solidification of the M–S–H cement does not lead to any high local concentration of radionuclides within a short time, and can effectively improve safety during accidental leakage.

## 5. Conclusions

The influence of Cs^+^ on the hydration process of the M–S–H cement was evaluated by strength and microstructural analysis. By evaluating the mechanical properties, leaching resistance and freeze–thaw resistance of the solidified body, the curing effect of the M–S–H cement on Cs^+^ was determined. In addition, the following conclusions were drawn:
M–S–H cement has excellent mechanical properties. The solidified body exhibits a compressive strength with 1.6 wt % Cs^+^ at 90 d reaching 28.33 MPa, which is about three times higher than the compressive strength of the sample specified in the national standard GB14569.1-2011.Adding Cs^+^ only slightly affects the reaction of the M–S–H cement, and no new substance is found.M–S–H cement has excellent resistance to leaching. Cs^+^ exhibit leaching rates of 5.34 × 10^−5^ and 4.69 × 10^−4^ cm/d when the solidified body is immersed in deionized water and simulated sea water for 42 d. At this time, the CLFs of the solidified body are 8.89 × 10^−3^ cm and 5.60 × 10^−2^ cm, which are considerably lower than the requirements of the nuclide’s 42 days’ leaching rate, and the CLF stipulated in GB14569.1-2011.The leaching rate and CLF of the M–S–H cement-solidified body are smaller than those of the OPC-solidified body at each leaching stage. The CLF of the OPC-solidified body is about 5–7 times higher than that of the M–S–H cement.The curing of Cs^+^ nuclide by M–S–H cement is mainly the physical encapsulation and adsorption of hydration products.The compressive strength, impact resistance, freeze–thaw resistance and other properties of the M–S–H cement meet the requirements of the national standard GB14569.1-2011.


## Figures and Tables

**Figure 1 materials-13-00146-f001:**
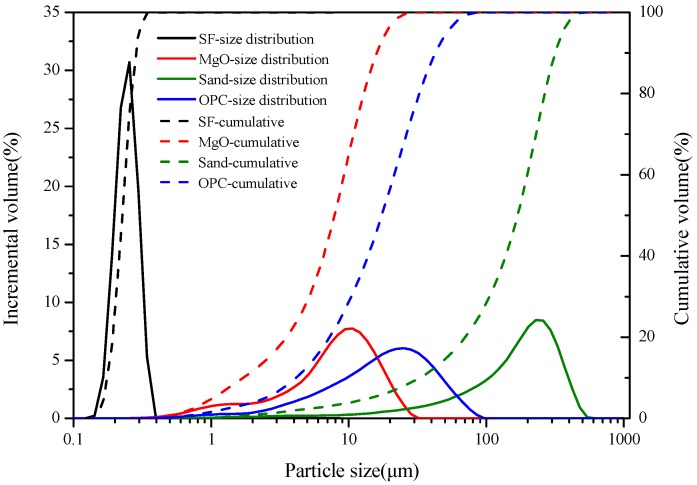
Particle size distribution of the main raw materials.

**Figure 2 materials-13-00146-f002:**
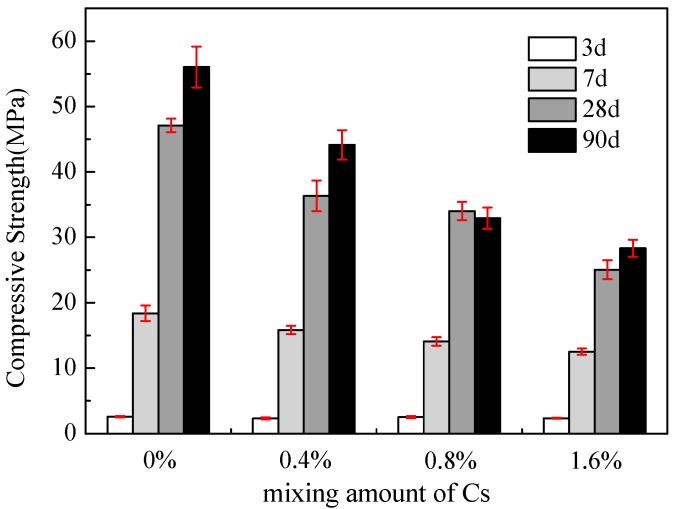
Compressive strength of M–S–H cement samples with different amounts of Cs^+^ after 3, 7, 28 and 90 d.

**Figure 3 materials-13-00146-f003:**
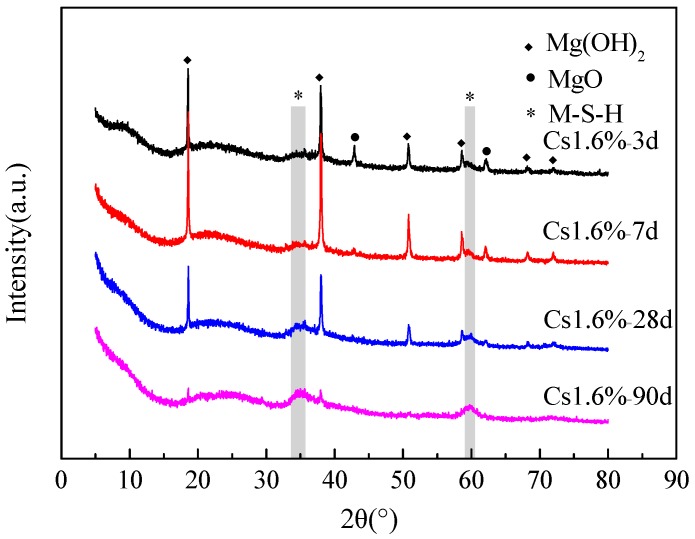
X-ray diffraction data of the M–S–H cement-solidified body with 1.6 wt % Cs^+^ after 3, 7, 28 and 90 d.

**Figure 4 materials-13-00146-f004:**
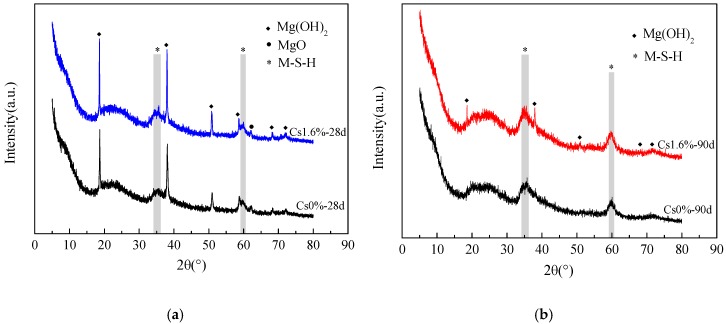
X-ray diffraction data of the M–S–H cement-solidified body with 1.6 wt % Cs^+^ and without Cs^+^ after (**a**) 28 and (**b**) 90 d.

**Figure 5 materials-13-00146-f005:**
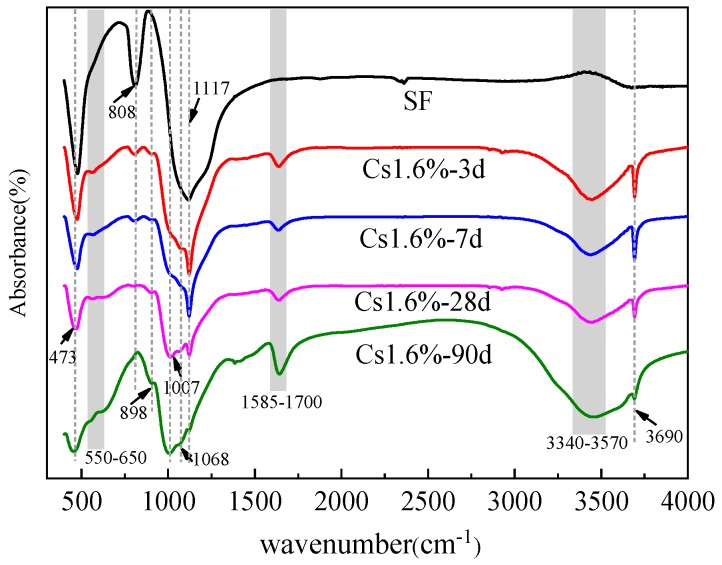
Fourier transform infrared spectra of the silica fume and M–S–H cement-solidified body with 1.6 wt % Cs^+^ after 3, 7, 28 and 90 d.

**Figure 6 materials-13-00146-f006:**
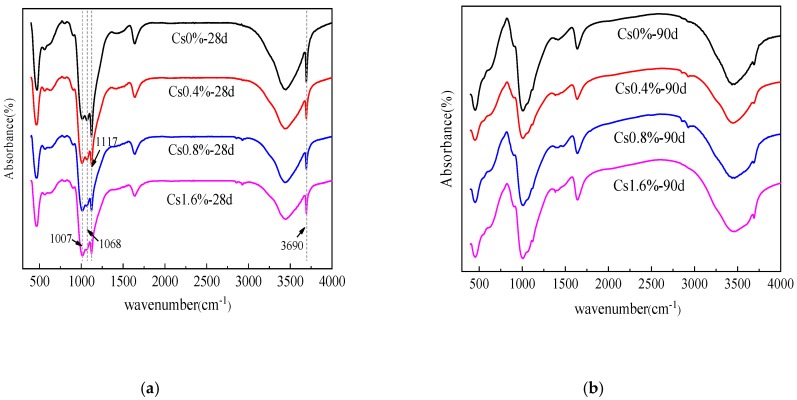
Fourier transform infrared spectra of the M–S–H cement-solidified body with 0 wt %, 0.4 wt %, 0.8 wt %, and 1.6 wt % Cs^+^ contents after (**a**) 28 and (**b**) 90 d.

**Figure 7 materials-13-00146-f007:**
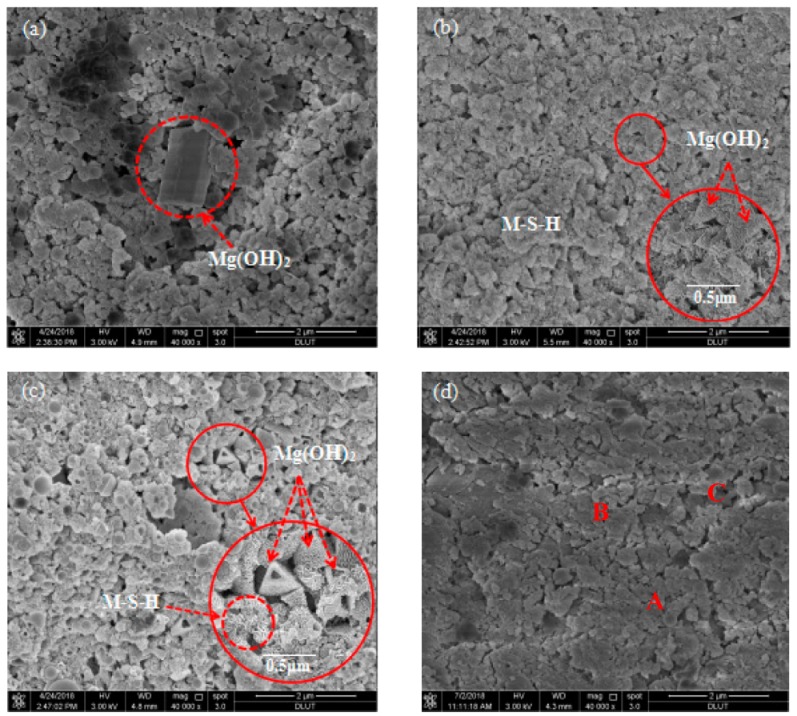
Microstructures of samples cured for different periods: (**a**) 3; (**b**) 7; (**c**) 28; and (**d**) 90 d.

**Figure 8 materials-13-00146-f008:**
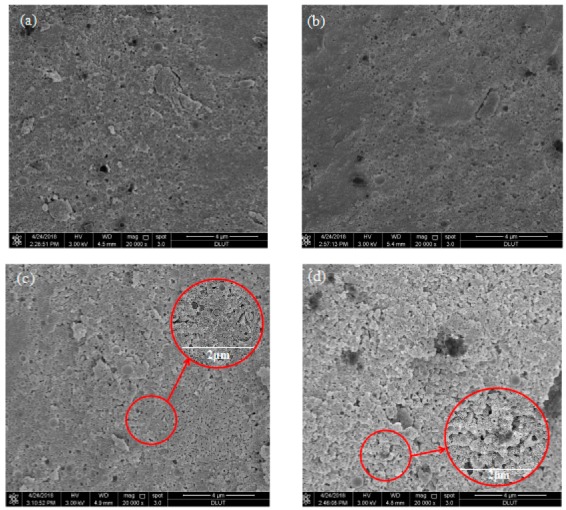
Scanning electron microscopy (SEM) of hydrated products with different Cs^+^ contents after 28 d: (**a**) 0%; (**b**) 0.4 wt %; (**c**) 0.8 wt %; and (**d**) 1.6 wt %.

**Figure 9 materials-13-00146-f009:**
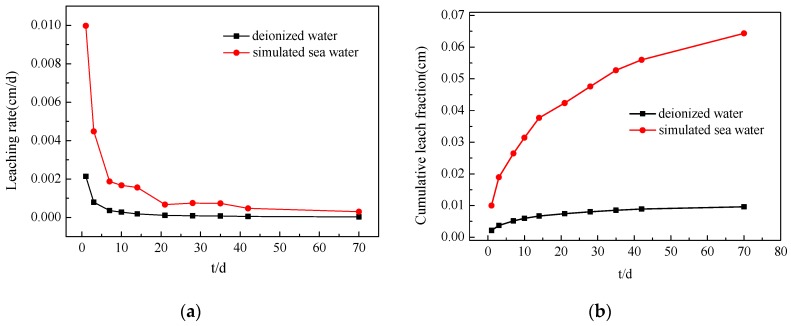
Leaching rate (**a**) and cumulative leach fraction (**b**) of the M–S–H cement-solidified body under different leaching fluids.

**Figure 10 materials-13-00146-f010:**
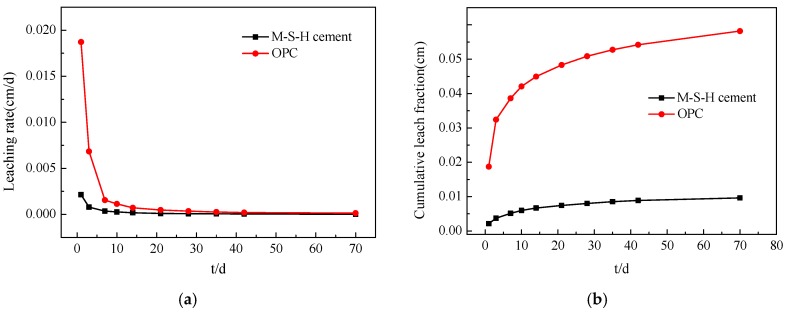
Leaching rate (**a**) and cumulative leach fraction (**b**) of different cement-solidified bodies.

**Figure 11 materials-13-00146-f011:**
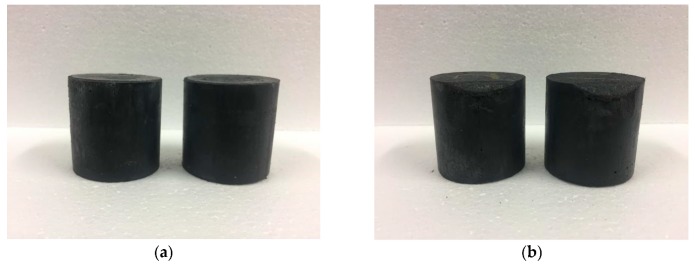
M–S–H cement-solidified body with 0.8 wt % Cs^+^ before (**a**) and after (**b**) anti-shock testing.

**Figure 12 materials-13-00146-f012:**
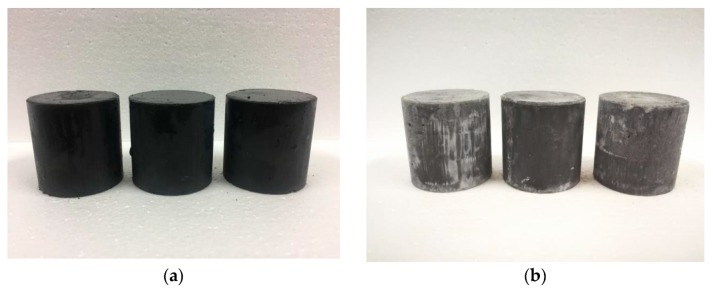
M–S–H cement-solidified body with 0.8 wt % Cs^+^ before (**a**) and after (**b**) freeze–thaw testing.

**Figure 13 materials-13-00146-f013:**
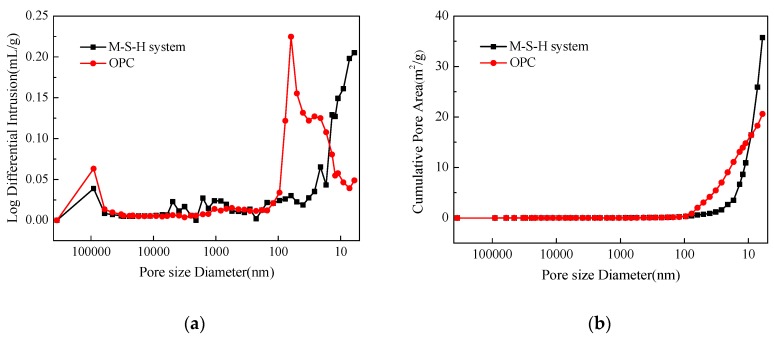
Pore size distribution (**a**) and accumulative pore area (**b**) of the M–S–H system-solidified body and ordinary Portland cement (OPC)-solidified body.

**Table 1 materials-13-00146-t001:** Characteristics of the raw materials.

Composition (wt %)	MgO	Silica Fume	OPC
MgO	98.451	0.452	2.08
SiO_2_	0.486	97.342	21.45
Al_2_O_3_	0.040	0.119	5.24
Fe_2_O_3_	0.179	0.046	2.89
CaO	0.616	0.428	61.13
MnO	0.037	0.039	-
SO_3_	-	0.353	2.50
P_2_O_5_	-	0.074	-
K_2_O	-	0.973	-
ZnO	-	0.029	-
Specific gravity (g/cm^3^)	3.23	1.94	3.1
Mean particle size (μm)	7.87	0.234	17.39
BET surface area (m^2^/g)	25	20.7	0.33

**Table 2 materials-13-00146-t002:** Synthetic seawater formula (Note: Diluted with water to 1000 g).

Component	NaCl	MgCl_2_	Na_2_SO_4_	CaCl_2_	KCl	NaHCO_3_	KBr
Quality (g)	23.497	4.981	3.917	1.102	0.664	0.192	0.096

**Table 3 materials-13-00146-t003:** Energy-dispersive X-ray spectroscopy (XDS) results of the M–S–H system-solidified body with 1.6 wt % Cs.

Element	A	B	C
	Atomic %	Atomic %	Atomic %
**O**	59.66	59.85	60.64
**Mg**	14.41	13.89	13.95
**Si**	17.30	15.18	17.19
**Cl**	0.10	0.11	0.12
**Cs**	0.17	0.16	0.17
**Mg:Si**	0.83	0.92	0.81

**Table 4 materials-13-00146-t004:** Leaching rate and cumulative leach fraction of the M–S–H cement-solidified body with 0.8 wt. % Cs under different leaching fluids.

Time/d	Deionized Water		Simulated Sea Water	
	Leaching Rate/(cm·d^−1^)	Cumulative Leach Fraction/cm	Leaching Rate/(cm·d^−1^)	Cumulative Leach Fraction/cm
1	2.14 × 10^−3^	0.00214	9.98 × 10^−3^	0.00998
3	7.90 × 10^−4^	0.00372	4.48 × 10^−3^	0.01894
7	3.57 × 10^−4^	0.00515	1.87 × 10^−3^	0.02643
10	2.74 × 10^−4^	0.00597	1.67 × 10^−3^	0.03144
14	1.83 × 10^−4^	0.00670	1.56 × 10^−3^	0.03767
21	1.04 × 10^−4^	0.00743	6.70 × 10^−4^	0.04236
28	8.52 × 10^−5^	0.00802	7.47 × 10^−4^	0.04759
35	7.10 × 10^−5^	0.00852	7.33 × 10^−4^	0.05272
42	5.34 × 10^−5^	0.00889	4.69 × 10^−4^	0.05600
70	2.55 × 10^−5^	0.00961	2.98 × 10^−4^	0.06434

**Table 5 materials-13-00146-t005:** Leaching rate and cumulative leach fraction of different cement-solidified bodies.

Time/d	M–S–H Cement		OPC	
	Leaching Rate/(cm·d^−1^)	Cumulative Leach Fraction/cm	Leaching Rate/(cm·d^−1^)	Cumulative Leach Fraction/cm
1	2.14 × 10^−3^	0.00214	1.87 × 10^−2^	0.01872
3	7.90 × 10^−4^	0.00372	6.84 × 10^−3^	0.03240
7	3.57 × 10^−4^	0.00515	1.56 × 10^−3^	0.03863
10	2.74 × 10^−4^	0.00597	1.15 × 10^−3^	0.04207
14	1.83 × 10^−4^	0.00670	7.15 × 10^−4^	0.04493
21	1.04 × 10^−4^	0.00743	4.82 × 10^−4^	0.04830
28	8.52 × 10^−5^	0.00802	3.65 × 10^−4^	0.05086
35	7.10 × 10^−5^	0.00852	2.70 × 10^−4^	0.05275
42	5.34 × 10^−5^	0.00889	2.07 × 10^−4^	0.05420
70	2.55 × 10^−5^	0.00961	1.42 × 10^−4^	0.05818

**Table 6 materials-13-00146-t006:** Strength and mass loss of the M–S–H cement-solidified body with 0.8 wt % Cs before and after freeze–thaw testing.

	Before Testing	After Testing	Loss rate/%
Quality	187.05 g	186.08 g	0.52%
Compressive strength	24.33 MPa	20.93 MPa	13.97%

**Table 7 materials-13-00146-t007:** Pore structure analysis of M–S–H system-solidified body and OPC solidified body.

Matrix	Average Pore Size/nm	Median Pore Diameter/nm	Total Pore Area/(m^2^/g)	Porosity/%
M–S–H system	17.70	8.70	35.72	24.98
OPC	38.00	18.50	20.61	32.27

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
