# Peer review of "Immobilization of Radionuclide 133Cs by Magnesium Silicate Hydrate Cement"

_materials, 2019, doi:10.3390/ma13010146_

Round 1
Reviewer 1 Report
Manuscript ID: materials-646558
Type of manuscript: Article
Title: Research on immobilization of radionuclide 133Cs by magnesium silicate hydrate (M-S-H) cement
Authors: Tingting Zhang, Jing Zou, Yimiao Li, Shiwei Zhi, Tong Li, Yuan Jia*, Chris Cheeseman
1. Please add the particle size distributions of both MgO and SF starting materials.
2. Please add Cs mapping image and some EDS analysis results for the points.
3. Please introduce the relevant comments and answers concerning the solidification/stabilization of Cs by M-S-H - chemically bonding or physically bonded ?
4. Generally, there is no relevant results (from the structural point of view – XRD, FT-IR etc...) on immobilization of Cs by magnesium silicate hydrate.
I recommend this work after major revision.
Author Response
Response to Reviewer 1 Comments
1. Please add the particle size distributions of both MgO and SF starting materials.
Thanks for your comment. We have added the particle size distribution of main raw materials. (Figure 1)
2. Please add Cs mapping image and some EDS analysis results for the points.
We have added some EDS analysis results for the points (Figure 7 and Table 3). However, the Cs mass is very low, so the mapping image can’t show the distribution of Cs effectively.
3. Please introduce the relevant comments and answers concerning the solidification/stabilization of Cs by M-S-H - chemically bonding or physically bonded ?
I agree with you. There are no other relevant literatures on the solidification/stabilization mechanism of Cs by M-S-H. The experiment results show that solidification/stabilization mechanism is due to the physical adsorption and encapsulation. In the future, we will focus on the solidification/stabilization mechanism research.
4. Generally, there is no relevant results (from the structural point of view–XRD, FT-IR etc...) on immobilization of Cs by magnesium silicate hydrate.
Thanks for your comment. The XRD and FT-IR data show the amount of Mg(OH)2 in the sample with Cs+ is more than that in the blank sample at 28 d and 90 d. The results indicate that the addition of Cs+ reduces the rate of transformation from Mg(OH)2 to the M-S-H gel. In addition, no new substance is found.
Reviewer 2 Report
This manuscript deals with an actual and sensitive theme: immobilization of cesium is not adequate in case of OPC due to its solubility, investigation of different matrices is necessary. in this study inactive cesium has been used to study the immobilization effect of M-S-H cement matrix. it can be accepted after minor revision:
- it should be clearly defined the background in abstract and introduction- it is really disturbing how authors mix the inactive and active cesium.
- it should be mentioned that cesium has several isotopes, not only the mentioned (and consequently mixed) two isotopes exist. please rework the introduction.
Author Response
Response to Reviewer 2 Comments
This manuscript deals with an actual and sensitive theme: immobilization of cesium is not adequate in case of OPC due to its solubility, investigation of different matrices is necessary. in this study inactive cesium has been used to study the immobilization effect of M-S-H cement matrix. It can be accepted after minor revision:
1. It should be clearly defined the background in abstract and introduction- it is really disturbing how authors mix the inactive and active cesium. it should be mentioned that cesium has several isotopes, not only the mentioned (and consequently mixed) two isotopes exist. please rework the introduction.
Thanks for your comment. We have defined the background in the first paragraph of the Introduction.
137Cs has a long half-life of 30.5 years; moreover, action Cs exhibits medium and high levels of radiotoxicity. It is highly active and typically exists in the form of ions. It can easily migrate in soil and water and cause radioactive pollution when entering an environment with groundwater [8-10]. Solidification is an important process of radioactive waste disposal. In the experiment, we often use inert Cs (e.g. CsCl) instead of active Cs in order to avoid the harm of radiotoxicity [6,7].
Reviewer 3 Report
Research on immobilization of radionuclide 133Cs by magnesium silicate hydrate (M-S-H) cement
The article is about synthesis of cement matrices for immobilization cesium, and about researching their physicochemical characteristics. The article has high scientific level with a large number of experimental dates and it will be of interest to specialists, working in processing and conditioning radioactive wastes.
Despite the high level of research, I have some questions, comments and remarks to authors.
1. Please, include in introduction this publication: Kononenko, O.A.; Gelis, V.M.; Milyutin, V.V. Incorporation of bottoms from nuclear power plants into a matrix based on portland cement and silicic additives. At Energy 2011, 109, 278–284.
2. Line 32-34. Authors write «There has been 360,000 tons of radioactive nuclear waste accumulated by 445 nuclear reactors in total, with the annual growth rate of 12,000 tons». What is the type of wastes, liquid or solid? Give the reference on this fact.
3. Line 36. Ions Cs+ do not have a noticeable toxic effect to organism, as example Cd and Please replace the term toxicity with radiotoxicity.
4. Line 127. Why distilled water and simulated sea water was used? Give an example, when will cement matrices contact with seawater in real conditions? Why is leaching not studied inground water?
5. Symbols on pictures 2 and 3 is poorly readable and barely distinguishable. Please, make the symbols good distinguishable.
6. When authors made cement matrices, they add cesium in pure form without additives i.e. the conditions are perfect. Explain, is there problem of immobilization of cesium in «purest form»? Are there dates that can show, that availability of another impurities, except cesium, does not influence on matrix strength? It is very important, because in real conditions radioactive wastes, that was cemented, can be in form of sludge (Osmanlioglu, A.E. Immobilization of radioactive waste by cementation with purified kaolin clay. Waste Management 2002, 22, 481–483), and also in form of solution with complex chemistry composition (Ojovan, M.I.; Varlackova, G.A.; Golubeva, Z.I.; Burlaka, O.N. Long-term field and laboratory leaching tests of cemented radioactive wastes. Journal of Hazardous Materials 2011, 187, 296–302). The excess of sodium, containing in that radioactive wastes, can lead to matrix strength reduction. It is important, because increase containing of Cs с 0.4wt% to 6wt%, leads to a decrease in the matrix strength by almost a factor of two. If the Authors do not have such data, I would recommend to slightly adjust the introduction and formulation of the research problem.
Author Response
Response to Reviewer 3 Comments
The article is about synthesis of cement matrices for immobilization cesium, and about researching their physicochemical characteristics. The article has high scientific level with a large number of experimental dates and it will be of interest to specialists, working in processing and conditioning radioactive wastes.
Despite the high level of research, I have some questions, comments and remarks to authors.
1. Please, include in introduction this publication: Kononenko, O.A.; Gelis, V.M.; Milyutin, V.V. Incorporation of bottoms from nuclear power plants into a matrix based on portland cement and silicic additives. At Energy, 2011, 109, 278–284.
Thanks for your comment. We have modified this. (the new text: line 62-64)
2. Line 32-34. Authors write «There has been 360,000 tons of radioactive nuclear waste accumulated by 445 nuclear reactors in total, with the annual growth rate of 12,000 tons». What is the type of wastes, liquid or solid? Give the reference on this fact.
We have modified this part.
There has been 360,000 tons of radioactive nuclear waste accumulated by 445 nuclear reactors in total, with the annual growth rate of 12,000 tons , mainly the radioactive waste liquid produced in the process of power generation [4-6]. (the new text: line 32-35)
3. Line 36. Ions Cs+ do not have a noticeable toxic effect to organism, as example Cd and Please replace the term toxicity with radiotoxicity.
We have modified this part. (the new text: line 39)
4. Line 127. Why distilled water and simulated sea water was used? Give an example, when will cement matrices contact with seawater in real conditions? Why is leaching not studied inground water?
Thanks for your comment. According to the standard test for leachability of low and intermediate level solidified radioactive waste forms (GB/T 7023-2011) and performance requirements for low and intermediate level radioactive waste form-cemented waste form (GB14569.1-2011), the leaching solution can choose distillation liquid or simulated sea water.
5. Symbols on pictures 2 and 3 is poorly readable and barely distinguishable. Please, make the symbols good distinguishable
We have modified Figure 2 and 3 to make the symbols more distinguishable.
6. When authors made cement matrices, they add cesium in pure form without additives i.e. the conditions are perfect. Explain, is there problem of immobilization of cesium in «purest form»? Are there dates that can show, that availability of another impurities, except cesium, does not influence on matrix strength? It is very important, because in real conditions radioactive wastes, that was cemented, can be in form of sludge (Osmanlioglu, A.E. Immobilization of radioactive waste by cementation with purified kaolin clay. Waste Management 2002, 22, 481–483), and also in form of solution with complex chemistry composition (Ojovan, M.I.; Varlackova, G.A.; Golubeva, Z.I.; Burlaka, O.N. Long-term field and laboratory leaching tests of cemented radioactive wastes. Journal of Hazardous Materials 2011, 187, 296–302). The excess of sodium, containing in that radioactive wastes, can lead to matrix strength reduction. It is important, because increase containing of Cs с 0.4wt% to 6wt%, leads to a decrease in the matrix strength by almost a factor of two. If the Authors do not have such data, I would recommend to slightly adjust the introduction and formulation of the research problem.
Thanks for your comment. I agree with you. The real conditions radioactive wastes, that were cemented, can be in form of sludge and also in form of solution with complex chemistry composition. In this paper, we studied the solidification/stabilization performance of Cs without considering the influence of other components (such as sludge and Ca(NO3)2/NaNO3). The solidification of Cs via MSH cement has not been reported. In the future, we will focus on the effect of other components in nuclear waste on the cementitious matrix.
Round 2
Reviewer 1 Report
I accept all received responses from the Authors.